# Exploring gender dysphoria and related outcomes in a prospective cohort study: protocol for the Swedish Gender Dysphoria Study (SKDS)

Fatih Özel,[1,2] Malin Indremo,[3] Georgios Karamanis,[3] Ulf Elofsson,[3,4] Ulrika Beckman,[5] Attila Fazekas,[6] Louise Frisén,[7] Magnus Isaksson,[8] Lotta Sandström,[9,10] Nils Thelin,[11] Åsa Tivesten,[12,13] Jeanette Wahlberg,[14] Alkistis Skalkidou,[4] Owe Bodlund,[10] Fotios C Papadopoulos [iD] [3]

**To cite:** Özel F, Indremo M, Karamanis G, *et al*. Exploring gender dysphoria and related outcomes in a prospective cohort study: protocol for the Swedish Gender Dysphoria Study (SKDS). *BMJ Open* 2023;13:e066571. doi:10.1136/bmjopen-2022-066571

For numbered affiliations see end of article.

**Correspondence to**
Dr Fotios C Papadopoulos;
fotis.papadopoulos@neuro.uu.se

## ABSTRACT

**Introduction** There has been a drastic increase in the reported number of people seeking help for gender dysphoria in many countries over the last two decades. Yet, our knowledge of gender dysphoria and related outcomes is restricted due to the lack of high-quality studies employing comprehensive approaches. This longitudinal study aims to enhance our knowledge of gender dysphoria; different aspects will be scrutinised, focusing primarily on the psychosocial and mental health outcomes, prognostic markers and, secondarily, on the underlying mechanisms for its origin.

**Methods and analysis** The Swedish Gender Dysphoria Study is an ongoing multicentre longitudinal cohort study with 501 registered participants with gender dysphoria who are 15 years old or older. Participants at different phases of their clinical evaluation process can enter the study, and the expected follow-up duration is three years. The study also includes a comparison group of 458 age- and county-matched individuals without gender dysphoria. Data on the core outcomes of the study, which are gender incongruence and experienced gender dysphoria, body satisfaction and satisfaction with gender-affirming treatments, as well as other relevant outcomes, including mental health, social functioning and life satisfaction, are collected via web surveys. Two different research visits, before and after starting on gender-affirming hormonal treatment (if applicable), are planned to collect respective biological and cognitive measures. Data analysis will be performed using appropriate biostatistical methods. A power analysis showed that the current sample size is big enough to analyse continuous and categorical outcomes, and participant recruitment will continue until December 2022.

**Ethics and dissemination** The ethical permission for this study was obtained from the Local Ethical Review Board in Uppsala, Sweden. Results of the study will be presented at national and international conferences and published in peer-reviewed journals. Dissemination will also be implemented through the Swedish Gender Dysphoria Study network in Sweden.

## STRENGTHS AND LIMITATIONS OF THIS STUDY

⇒ The longitudinal design with a large sample size is the biggest strength of this multicentre cohort study, considering that gender dysphoria is an uncommon diagnosis.
⇒ Having a comparison group consisting of individuals without gender dysphoria stands as a valuable advantage.
⇒ A comprehensive assessment of psychosocial determinants is being performed.
⇒ Participants enter the study at different stages of their clinical evaluation process, which partly results in missing baseline information.
⇒ The representativeness of the target population could be affected by the fact that persons with more mental health issues might tend to participate less often or drop out in higher extent.

## INTRODUCTION

Gender dysphoria (GD) is, according to the latest version of the Diagnostic and Statistical Manual of Mental Disorders (5th Edition),[1] a condition that involves distress due to incongruence between an individual's birth-assigned sex and gender identity. Persons with GD can identify themselves within binary and non-binary gender identities; non-binary identities may refer to having different gender identities simultaneously, having a neutral or fluid gender identity, not having a gender identity precisely, and so forth.[2] During the last decades, there has been an increasing number of individuals with GD seeking gender-affirming healthcare in many countries.[3 4] This increase is more pronounced among the younger age groups, and is more noticeable among individuals who are assigned female at birth (AFAB).[5] Similar rising patterns for GD have also been demonstrated in Sweden recently.

An increase in the incidence of GD was observed during the period 2004–2015 in Sweden, with the most profound increase among AFAB in the age group 10–17 (20%–30% average yearly increase), while 10–17 year-old assigned male at birth (AMAB) and both AFAB and AMAB in the age group 18–30 years had an average yearly increase of 10%–20%.[6] In the latest report of the National Board of Health and Welfare, 0.06% of the Swedish population was stated to have a GD diagnosis in 2018.[7] Even though an increasing trend in GD and the consequent need for gender-affirming healthcare is widely recognised, our understanding of GD and gender-affirming healthcare is substantially limited in many aspects.

The origin of GD is unknown but assumably multifactorial, with biological factors interacting with the psychosocial environment. There are established sex differences in the brain, concerning neuroanatomy, neurochemistry as well as neurocognition.[8] One hypothesis for the emergence of GD is that the processes of sexual differentiation of the genitals and the brain may be incongruent.[8] However, evidence from multiple neuroimaging studies has not given support to this, and alternative hypotheses have been proposed.[9 10] Genetic factors have also been observed to play a role in GD, and twin studies have been mostly employed to enlighten the development in GD.[11–13] Even though the existing evidence remains largely inconclusive, genetic factors seem to be involved through altered sex hormone signalling; the oestrogen and androgen receptors seem to be implicated in the genetic basis of GD.[14] Moreover, epigenetic alterations among individuals with GD have been recently proposed,[15] supporting evidence highlighting the importance of the interaction between genetic architecture and environment in the development of GD. Thus, early-life environmental exposures could potentially be associated with the biological underpinning of GD via epigenetic modifications. Yet, the current evidence on biological determinants of GD is inconclusive in the literature.

An important aspect in the field of GD research relates to the mental health of individuals with GD. A growing body of evidence shows that persons with GD are more vulnerable to mental illnesses. A recent study using Swedish population register data has reported that mood and anxiety disorders, use of antidepressant or anxiolytic medication, and hospitalisation rates due to suicide attempts are more common among persons with GD in comparison to the general population.[16] In addition to psychiatric diagnoses, transgender persons' quality of life is radically affected, with reported lower life satisfaction.[17] The aforementioned adverse consequences on the mental well-being of individuals with GD could be explained through experienced dysphoria and stigma-related stress. This framework of stigmatisation and discrimination in this context is conventionally called minority stress and was established almost 20 years ago.[18] According to the minority stress model, gender minority groups endure particular stressors such as discrimination, lack of social support, and violence, and this intense exposure to stressors could result in mental health issues. Individuals with GD are unfortunately no exception to minority gender groups, and their health is influenced by different sources of stigma-related stress.[17 19] More research is required on the experience of discrimination and violence by persons with GD and its effects on their mental health.

Gender-affirming medical treatment is an additional dimension that needs to be addressed in research. Treatment guidelines are provided in the World Professional Association for Transgender Health's Standards of Care, currently in its eighth version.[20] Depending on individual needs and wishes, treatments involve gender-affirming hormonal treatment (GAHT), gender-affirming surgical procedures, consultations and potential interventions for reproductive health, voice therapy, hair removal and use of prostheses. GAHT consists mainly of oestrogens and antiandrogens for AMAB and testosterone for AFAB. The goal of GAHT is to induce the development of bodily characteristics congruent with the self-identified gender, thereby improving mental health and quality of life. For children and adolescents, puberty suppression with gonadotropin-releasing hormone analogues is implemented to delay the development of secondary sexual characteristics. The intention is to alleviate the distress associated with the development of secondary sex characteristics, thereby providing a time for ongoing discussion and exploration of gender identity before deciding whether to take more irreversible steps. Previous studies posit that GAHT ameliorates the symptoms of GD, increases the quality of life, alleviates depression, anxiety and suicidality, and improves psychosocial functioning among both young and adult individuals.[21–24] One should still acknowledge the limitations of interpreting the results from previous research. Moreover, the reports on the potential adverse health outcomes related to GAHT make research in this dimension even more indispensable.[25 26]

There has been an increase in research studies related to GD in parallel to the increase in GD prevalence. Still, we lack evidence focused on some particular aspects of GD, such as biological and psychological determinants, mental health and psychosocial outcomes, and the impact of gender-affirming treatment (GAT) on these outcomes. The majority of the studies conducted in the field of GD have methodological limitations, with small sample sizes, cross-sectional design, and no comparison groups. Larger longitudinal studies are required to improve our understanding of GD and associated outcomes. More research is especially warranted among adolescents and young adults with GD, as those are the age groups with the most profound incidence increase during the last decade.

The Swedish Gender Dysphoria Study (Svenska Könsdysforistudien, SKDS) thus primarily aims to investigate (1) mental health outcomes associated with GD, for instance, depressive and anxiety symptoms and suicidality, (2) psychosocial outcomes in relation to GD, such as quality of life, global functioning, and social vulnerability

(along with support, discrimination and violence), (3) different aspects of GAT, including the impact on mental health and psychosocial outcomes and, secondarily, (4) biological correlates of GD containing the effects of early-life environmental exposures, genetic underpinnings, epigenetic alterations, and potential prognostic biological markers.

## METHODS AND ANALYSIS
### Study design and participants
SKDS (www.skds.se) is an ongoing multicentre longitudinal cohort study in Sweden, which was established in 2016. All participants in contact with one of the associated GD evaluation clinics are eligible to be asked to participate. Having contact with a clinic for GD, being at least 15 years old, and speaking Swedish are set as the inclusion criteria. Relevant information about the study is sent to all participants and written informed consent is obtained. The right of leaving the study without any obligation to report a reason for that and the fact that participating or not in the study has no effect on the received healthcare are clearly indicated during the informed consent process. Separate informed consents are asked for different parts of the study (eg, participating in surveys, research visits, online cognitive tests, giving biological samples, etc). Recruitment of participants is ongoing in the GD clinics in Alingsås/Gothenburg, Linköping, Lund/Malmö, Stockholm (centre for evaluation of children), Umeå, and Uppsala. In Sweden, the evaluation process and treatment for GD is centralised to these six regional clinics working in multidisciplinary teams, all following a national consensus programme.

The participants of SKDS are followed up longitudinally from inclusion to three years later, with several assessments of variables of interest. Due to the naturalistic design of the study, the participants can be at different phases in their clinical evaluation process when they enter the study. The recruitment of the participants is expected to be finished in December 2022; therefore, the follow-up is predicted to be finalised in December 2025.

There are currently 501 registered participants with GD in SKDS. In January 2019, the governmental agency Statistics Sweden provided 30 random controls with no registered diagnosis of GD up to that time (15 with male legal gender and 15 with female legal gender) for each participant in the study. To recruit controls, an invitation letter and one reminder were sent via mail to 8790 persons in total; a comparison group thus was constituted with 458 age- and county-matched controls without a diagnosis of GD. The controls are only assessed once with the same baseline instruments as the participants with GD. Regarding the low participation rate for the controls, the dropout analysis conducted by Statistics Sweden showed higher dropout rates for controls with male legal gender, those born outside Sweden, and those without Swedish citizenship. No other major biases were observed for age, income, or municipality. The second round of

recruitment of controls will be executed at the beginning of 2023, in order to acquire more controls for recruited participants since 2019.

In order to plan and follow the study coordination and progress as well as the utilisation of all collected data, a national steering committee for the SKDS was created in November 2015, consisting of clinicians and researchers in the field of psychiatry and endocrinology from the collaborating clinics. Representatives from interest organisations, including Transammans and RFSL Ungdom in Stockholm and Uppsala, have an advisory role in the SKDS.

### Data collection and measures
#### Main outcomes and psychosocial measures
Longitudinal web surveys are the main tools for data collection in the current study. Sociodemographic characteristics and outcomes of relevance for GD are measured through previously validated instruments listed in table 1, as well as three specific surveys for SKDS, Surveys I–III, described in table 2. The core outcomes of the current study are gender incongruence and experienced GD, body satisfaction and satisfaction with GAT. Assessment of the satisfaction with each of the received GATs, as well as overall satisfaction with GAT, is covered in depth in the last SKDS-specific survey. For each of the received treatments, the participants are asked the following questions: 'how satisfied are you with the physical result of treatment x?', 'how has treatment x affected your mental well-being?' and 'if you were to decide on initiating treatment x today, would you make the same decision again?'.

In addition to the core outcomes, the included instruments cover general mental health status, previous and current psychiatric diagnoses, traits of autism spectrum disorder and attention deficit hyperactivity disorder, alcohol and substance use, social functioning, and life satisfaction. Furthermore, a scale on potential side effects of treatment is administered after the initiation of GAT.

#### Cognitive measures
All participants who have not started on GAHT are invited to undertake a computer-based neurocognitive assessment battery, called WebNeuro, which can be administered online or with a local installation.[27] Different domains are addressed within the battery, including sensorimotor, memory, executive functioning, attention, and social cognition. Somatomap, an online tool for assessing body perception, is also administered to all participants.[28]

Participants recruited from the GD evaluation clinic in Uppsala are also invited to a research visit at the Uppsala University Hospital where they are asked to undertake specific neurocognitive tests called Cambridge Gambling Task and Stockings of Cambridge from the CANTAB battery.[29] All cognitive tests are performed at baseline, before GAHT, and at least 6 months after start of GAHT.



**Table 1** Outline of the validated questionnaires used during different time points

| Variables of interest | Questionnaires | M1 | M3 | M6 | M12 | M18–M19 | M24–M25 | M36–M37 |
|---|---|---|---|---|---|---|---|---|
| GD symptoms | TCS[30] | ✔ | ✔ | ✔ | ✔ | ✔ | ✔ | ✔ |
| Body satisfaction | BIS[31] | ✔ | | | | ✔ | | ✔ |
| Gender identity and roles during childhood | RCGI[32] | ✔ | | | | | | |
| Life satisfaction | GCLS[33] | ✔ | | ✔ | ✔ | ✔ | ✔ | ✔ |
| Mental health symptoms | DSM-5 screening[34] | ✔ | ✔ | ✔ | ✔ | ✔ | ✔ | ✔ |
| Autistic features | RAADS-14[35] AQ[36] | ✔ | | | | | | |
| ADHD characteristics | ASRS[37] WURS[38] | ✔ | | | | | | |
| Alcohol use | AUDIT-C[39] | ✔ | | | | ✔ | | ✔ |
| Substance use | DUDIT-C[40] | ✔ | | | | ✔ | | ✔ |
| Gaming addiction | GAS[41] | ✔ | | | | ✔ | | ✔ |
| Gambling screening | NODS-CLiP[42] | ✔ | | | | ✔ | | ✔ |
| Eating disorder symptoms | EDE-Q[43] | ✔ | | | | ✔ | | ✔ |
| Personality traits | SSP[44] | ✔ | | | | | ✔ | |
| Functional impairment and disability | SDS[45] WHODAS 2.0[46] | ✔ | | | | ✔ | | ✔ |

ADHD, attention deficit hyperactivity disorder; AQ, Autism Spectrum Quotient; ASRS, Adult ADHD Self-Report Scale; AUDIT-C, Alcohol Use Disorders Identification Test-Consumption; BIS, Body Image Scale; DSM-5 screening, Diagnostic and Statistical Manual of Mental Disorders 5th edition Self-Rated Level 1 Cross-Cutting Symptom Measure; DUDIT-C, Drug Use Disorders Identification Test-Consumption; EDE-Q, Eating Disorder Examination Questionnaire; GAS, Game Addiction Scale; GCLS, Gender Congruence and Life Satisfaction Scale; GD, gender dysphoria; M, month; NODS-CLiP, Loss of Control, Lying, and Preoccupation Scale; RAADS-14, Ritvo Autism and Asperger Diagnostic Scale 14; RCGI, Recalled Childhood Gender Identity/Gender Role Questionnaire; SDS, Sheehan Disability Scale; SSP, Swedish Universities Scales of Personality; TCS, Transgender Congruence Scale; WHODAS 2.0, WHO Disability Assessment Schedule 2.0; WURS, Wender Utah Rating Scale.

## Biological measures

For genetic and epigenetic analyses, all participants are asked to contribute with saliva samples, deciduous teeth, and the consent to use newborn dried blood spots which are stored in the national PKU-biobank. Saliva samples will also be collected after a period of at least 6 months after starting on GAHT for individuals without GAHT at baseline in order to do epigenetic analyses related to the GAHT.

During their research visits, participants recruited from the GD evaluation clinic in Uppsala are also invited to contribute with more biological measures before and after the initiation of GAHT. The measures consist of blood samples to analyse hormonal levels and possible changes in the immune system, a physical examination including blood pressure, weight and hip-waist ratio as well as measuring bioimpedance and heart rate variability. The first research visit is conducted during baseline assessment and a follow-up visit at least 6 months after initiation of GAHT.

These biological samples will be used in order to study the potential biological correlates of GD consisting of the effects of early-life environmental exposures, genetic underpinnings, epigenetic alterations as well as potential prognostic biomarkers.

## Electronic health records and national register data

Information related to participants' GD evaluation can be retrieved from the electronic health records for those who have given their consent. Similarly, for those with consent, a linkage will be made to the Swedish national registers, such as the National Patient Register at the National Board of Health and Welfare and Statistics Sweden. These two information sources will be used to supplement the information collected with the web surveys.

## Statistical analysis

The research team of SKDS will perform the statistical analyses using the most recent version of the statistical software R. We will manage different types of missing data with different approaches. First, a potential bias that could arise from loss to follow-up will be evaluated; characteristics of missing participants will be addressed with dropout analysis and presented. Second, missing data and loss to follow-up will be diminished with chart review of clinical records and linkage to national health registers. We will also execute multiple imputation where it is required, and pooled results will be presented from imputed data sets.

Initially, summary statistics, including the descriptive characteristics of participants will be provided.

**Table 2** Outline of the SKDS-specific questionnaires used during different time points

| Category | Survey I Month 1 | Survey II Months 3, 6, 12, 18, 24 | Survey III Month 36 |
|---|---|---|---|
| **Background information** | | | |
| Age | ✔ | | |
| Birth-assigned gender | ✔ | ✔ | ✔ |
| Education | ✔ | | ✔ |
| Family | ✔ | | ✔ |
| Occupational status | ✔ | | ✔ |
| Marital/relationship status | ✔ | | ✔ |
| Current contact with gender identity clinic | ✔ | ✔ | ✔ |
| **Gender identity** | | | |
| Gender identity | ✔ | ✔ | ✔ |
| Preferred pronoun(s) | ✔ | ✔ | ✔ |
| Openness about gender identity | ✔ | ✔ | ✔ |
| **Gender dysphoria** | | | |
| Perceived distress from GD | ✔ | ✔ | ✔ |
| Body satisfaction | ✔ | ✔ | ✔ |
| Preferences regarding GAT | ✔ | | ✔ |
| **Gender-affirming treatments (GAT)** | | | |
| Hormonal treatments | ✔ | ✔ | ✔ |
| Top surgeries | ✔ | ✔ | ✔ |
| Bottom surgeries | ✔ | ✔ | ✔ |
| Speech therapy | ✔ | ✔ | ✔ |
| Fertility preservation | ✔ | ✔ | ✔ |
| Satisfaction with each of the treatments | | | ✔ |
| Overall satisfaction with GAT | | | ✔ |
| **Minority stress and support** | | | |
| Experiences of discrimination and violence | | ✔ | |
| Perceived social support | | ✔ | |
| **Sexuality** | | | |
| Sexual identity | ✔ | | ✔ |
| Sexual practice | ✔ | | ✔ |
| Satisfaction | ✔ | | ✔ |
| **Mental health** | | | |
| Mental health history | ✔ | | |
| Current psychiatric diagnoses | ✔ | | ✔ |
| Experiences of trauma | ✔ | | ✔ |
| Self-harm | ✔ | | ✔ |
| Suicidality | ✔ | | ✔ |
| Perceived overall mental health status | ✔ | ✔ | ✔ |

Continued

**Table 2** Continued

| Category | Survey I Month 1 | Survey II Months 3, 6, 12, 18, 24 | Survey III Month 36 |
|---|---|---|---|
| **Physical health** | | | |
| Chronic illness history | ✔ | | ✔ |

GD, gender dysphoria; SKDS, Svenska Könsdysforistudien.

Cross-sectional and longitudinal relationships will be assessed using the distinctive outcomes we mentioned above. Logistic regression analysis will be used for the categorical outcomes, such as having been diagnosed with GD or not. Continuous outcomes will be analysed with linear regression analysis. For longitudinal relations, we will conduct linear mixed models with repeated measures. We will use directed acyclic graphs to identify confounders and mediators. Confounders will be adjusted in different models, whereas potential mediators will also be assessed with mediation analysis. We will apply multiple testing correction wherever necessary.

Regarding statistical power analysis, we need data for 52 participants of each birth-assigned sex in order to detect a 25% increase from a mean Transgender Congruence Scale score of 2.1, with alpha=0.05 and power=90%, using a Mann-Whitney U test for the outcome of GD. Concerning the categorical mental health outcomes, expecting a change from 30% to 15% (as we have seen for the prevalence of self-harm thoughts in preliminary analysis from the SKDS data), we need 161 participants for alpha=0.05 and power=90%, using a $\chi^2$ test. For a smaller change from 30% to 20%, we need 294 participants for alpha=0.05 and power=80%. The participation rate for SKDS is approximately 70%. Among those participants, approximately 70% actively participate in the majority of our surveys. The expected final number of recruited participants in the study is around 600, which leaves nearly 400 actively engaged participants.

## Patient and public involvement

The study was initially designed by the research group comprising clinicians and researchers working clinically with individuals with GD. During the recruitment phase a collaboration was established with representatives from interest organisations in Sweden, which are RFSL Ungdom and Transammans; this collaboration is ongoing with regular meetings. Persons with GD were involved in the design of SKDS through interest organisation representatives; the researchers were informed by the representatives concerning the contemporary needs of individuals with GD, and the research questions; methods and outcome measures were adjusted and further developed based on the feedback received. Interest organisations will be asked to contribute to the dissemination of the results through their networks. The results will also be disseminated through the project's website which is publicly accessible.



## ETHICS AND DISSEMINATION

Ethical awareness has critical importance in clinical studies since patients might feel pressured to participate to access healthcare services. In SKDS, the voluntary participation and the fact that participation has no effect whatsoever on received care are clarified both orally and in writing. Study participants were also informed that they could cancel their participation without any justification at any stage of follow-up. Furthermore, data breach might become an important risk, as sensitive information at the individual level is stored. However, we have taken all measures, such as pseudonymisation, storage of the data on secure servers at Uppsala University, and separating the personal identifiers (ie, name and personal identity number) from the self-reports and other collected data. The study was approved by the Ethical Review Board in Uppsala (Dnr: 2016/013).

We plan to disseminate the research findings from the SKDS through publication in peer-reviewed journals and presentations at relevant national and international conferences as well as at dedicated seminars or workshops organised by the involved researchers, which could be also of interest to governmental and non-governmental bodies. Dissemination will be assured through the project's website (www.skds.se) as well as social media platforms. The results will be delivered to all clinical teams working with GD evaluation in Sweden through direct communication within the SKDS network. Contact will also be initiated with the national societies of general physicians, psychiatrists, psychologists, endocrinologists as well as obstetricians and gynaecologists in Sweden.

**Author affiliations**
[1]Department of Organismal Biology, Uppsala University, Uppsala, Sweden
[2]Centre for Women's Mental Health during the Reproductive Lifespan (WOMHER), Uppsala University, Uppsala, Sweden
[3]Department of Medical Sciences, Psychiatry, Uppsala University, Uppsala, Sweden
[4]Department of Women's and Children's Health, Uppsala University, Uppsala, Sweden
[5]Department of Psychiatry, Sahlgrenska University Hospital, Gothenburg, Sweden
[6]Department of Psychiatry, Lund University, Lund, Sweden
[7]Department of Clinical Neuroscience, Karolinska Institute, Stockholm, Sweden
[8]Department of Medical Sciences, Endocrinology and Mineral Metabolism, Uppsala University, Uppsala, Sweden
[9]ANOVA, Andrology, Sexual Medicine and Transgender Medicine, Karolinska University Hospital, Stockholm, Sweden
[10]Department of Clinical Sciences, Psychiatry, Umeå Universitet, Umeå, Sweden
[11]Department of Psychiatry, Linköping University Hospital, Linköping, Sweden
[12]Wallenberg Laboratory for Cardiovascular and Metabolic Research, Department of Molecular and Clinical Medicine, Institute of Medicine, Sahlgrenska Academy, University of Gothenburg, Gothenburg, Sweden
[13]Department of Endocrinology, Sahlgrenska University Hospital, Gothenburg, Sweden
[14]Department of Endocrinology, Faculty of Medicine and Health, Örebro University, Örebro, Sweden

**Acknowledgements** We express our gratitude to all the study participants and the clinicians at the collaborating clinics. Moreover, we acknowledge Lucas Blixt and Vierge Hård from RFSL Ungdom Uppsala and Tom Summanen from Transammans for their advisory roles in the study.

**Contributors** AF, AS, JW, LF, LS, MIs, NT, OB, UB, ÅT and FCP were responsible for the conception and design of the study. FÖ drafted the protocol with significant contributions from MIn, GK, UE and FCP. All authors have critically revised the draft for intellectual content and approved the final manuscript. FCP is the principal investigator of the study.

**Funding** This work is supported by Region Uppsala, the Foundation of Nicke and Märta Nasvell, FORTE (Dnr: 2021-01968) and Centre for Women's Mental Health during the Reproductive Lifespan (WOMHER).

**Competing interests** None declared.

**Patient and public involvement** Patients and/or the public were involved in the design, or conduct, or reporting, or dissemination plans of this research. Refer to the Methods section for further details.

**Patient consent for publication** Not applicable.

**Provenance and peer review** Not commissioned; externally peer reviewed.

**ORCID iD**
Fotios C Papadopoulos http://orcid.org/0000-0002-8692-3652

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
