## [Reviewer comments · BMJ Open]

ARTICLE DETAILS

TITLE (PROVISIONAL)	Exploring gender dysphoria and related outcomes in a prospective cohort study: protocol for the Swedish Gender Dysphoria Study (SKDS)
AUTHORS	Özel, Fatih; Indremo, Malin; Karamanis, Georgios; Elofsson, Ulf; Beckman, Ulrika; Fazekas, Attila; Frisé, Louise; Isaksson, Magnus; Sandström, Lotta; Thelin, Nils; Tivesten, Åsa; Wahlberg, Jeanette; Skalkidou, Alkistis; Bodlund, Owe; Papadopoulos, Fotios C.

VERSION 1 – REVIEW

REVIEWER	Nat Thorne Nottingham Centre for Transgender Health , Nottingham Centre for Transgender Health
REVIEW RETURNED	17-Oct-2022

GENERAL COMMENTS	An interesting proposal for a study to examine both the origins and mental health impacts of gender dysphoria. Introduction I think there is a lack of description relating to the nature of the trans community here. it would be good to see a few sentences explaining how dysphoria affects trans people and what are the treatments used. It would also be good to break down identities and explain how some are binary and others are non-binary. I think the information about the possible origins of dysphoria is interesting, but doesn't seem to link well with what you are actually going to be testing in this area. I do feel as though the question of the origins of dysphoria is a totally separate question to mental health outcomes and wonder if it would be best as two distinct studies? Page 6, line 15 cites the SOC 7, but be aware the SOC 8 has now been published so 7 is not the latest version, as stated here. The aims do feel a bit far-reaching and disjointed. As I have stated, the early-life environmental exposures and genetic underpinnings and other biological aspects feel odd nestled within this study and warrants a study of its own I feel. Methods I feel the information on the qualitative part of the study is very much missing in the methods. I would like to know how the interviews are structured as well as what theoretical stance you are taking and how the data is being processed. Patient and Public Involvement Can you define further which interest organisations are taking part?
--

	In addition, are any trans people involved in the PPI? Ethics and Dissemination Can you confirm when the study will be completed by the participants and where? If it is presented to the patient at their initial assessment this could perhaps have an influence on the results, even though the participants are informed it does not form part of their assessment. My main issue with the research is that I feel the biological basis for
--	--

REVIEWER	Els Elaut Ghent University, Department of experimental clinical and health psychology
REVIEW RETURNED	24-Oct-2022

GENERAL COMMENTS	The current manuscript describes the Swedish Gender Dysphoria Study, an ongoing longitudinal cohort study (since 2016) among participants with gender incongruence over 15 years and older. Recruitment is still taking place in six gender teams in Sweden. Unique for this study is not only a web survey (assessing psychosocial and mental health outcomes and prognostic markers) is used, but also biological and cognitive data are collected. A matched control group was recruited. A secondary aim is to study the underlying mechanisms of gender dysphoria. The authors have written a very clear and comprehensive study description and are complemented with this very thorough and at the same time broad, biopsychosocial approach. A significant amount of participants could be included, and is expected to increase during the last phase of recruitment. I only have a few very minor suggestions to improve the manuscript. 1/ In the abstract, the authors mention an increased prevalence of mental issues in the limitations. Could they provide a reference for this sentence? 2/ In the meantime, WPATH has released the 8th version of the Standards of Care. Hence, I would suggest to update the reference on p. 6, second paragraph. 3/ On p.8 rule 43, the authors refer to several assessment times: baseline, before gender affirming hormonal treatment, and at least six months after. Could the authors clarify what those six months after mean: six months after the start of hormonal treatment? And why was this at least six months after, what is the maximum time since the start of hormonal treatment upon this assessment time? 4/ On p. 9, the authors describe how they ask the participants to consent to have a relative contacted for another study on the experiences of relatives on processes in transition-related health care. What instructions are being given on the participants to prevent both positive and negative selection bias in this other study? 5/ Also on p.9: could the authors explain the abbreviation 'TCS' when used for the first time?
---

VERSION 1 – AUTHOR RESPONSE

Reviewer: 1

Dr. Nat Thorne, Nottingham Centre for Transgender Health

Comments to the Author:

An interesting proposal for a study to examine both the origins and mental health impacts of gender dysphoria.

Thank you for reviewing our study protocol and for constructive comments to improve it.

Introduction

I think there is a lack of description relating to the nature of the trans community here. it would be good to see a few sentences explaining how dysphoria affects trans people and what are the treatments used. It would also be good to break down identities and explain how some are binary and others are non-binary.

Thank you for your comment and suggestions. We have now included the following text about binary and non-binary identities at the beginning of the introduction:

“Persons with GD can identify themselves within binary and non-binary gender identities; non-binary identities may refer to having different gender identities simultaneously, having a neutral or fluid gender identity, not having a gender identity precisely and so forth.” (Introduction, Page 3)

We provided an overview about mental health and gender dysphoria in the third paragraph of the introduction. In line with your suggestion, we elaborated on that part to make it clear:

*“The aforementioned adverse consequences on the mental well-being of individuals with GD could be explained through experienced dysphoria and stigma-related stress. This framework of stigmatization and discrimination in this context is conventionally called minority stress and was established almost 20 years ago.”*¹⁸ (Introduction, Page 4)

We summarized the treatments used in the fourth paragraph of the introduction as you suggested; the rest of the paragraph is still focused on the hormonal treatment as this will be specifically investigated in the SKDS.

“Depending on individual needs and wishes, treatments involves gender-affirming hormonal treatment (GAHT), gender-affirming surgical procedures, consultations and potential interventions for reproductive health, voice therapy, hair removal and use of prostheses.” (Introduction, Page 4)

I think the information about the possible origins of dysphoria is interesting, but doesn't seem to link well with what you are actually going to be testing in this area. I do feel as though the question of the origins of dysphoria is a totally separate question to mental health outcomes and wonder if it would be best as two distinct studies?

Thank you for this comment. We agree that the information about biological aspects in the introduction does not completely align with what we aim to investigate. Therefore, we updated the introduction and put more emphasis on genetic and epigenetic factors rather than cerebral circuits.

*“Genetic factors have also been observed to play a role in GD, and twin studies have been mostly employed to enlighten the development in GD.”*¹¹⁻¹³ *Even though the existing evidence remains largely inconclusive, genetic factors seem to be involved through altered sex hormone signaling; the estrogen and androgen receptors seem to be implicated in the genetic basis of GD.”*¹⁴ *Moreover, epigenetic alterations among individuals with GD have been recently proposed,¹⁵ supporting evidence highlighting the importance of the interaction between genetic architecture and environment in the development of GD. Thus, early-life environmental exposures could potentially be associated with the biological underpinning of GD via epigenetic modifications.”* (Introduction, Page 3)

We also agree that mental health outcomes and biological aspects could be investigated separately, and they are not necessarily interconnected. However, the aim of the SKDS is to improve our understanding of gender dysphoria at different levels; thus, there are various research questions under the main theme. We hope this will help us better understand the phenomenon and improve the support to persons with gender dysphoria as much as possible. Unfortunately, we cannot change the design of the study and main research questions at this phase as the study is ongoing.

Page 6, line 15 cites the SOC 7, but be aware the SOC 8 has now been published so 7 is not the latest version, as stated here.

Thank you for this comment. We updated the reference:

“Treatment guidelines are provided in the World Professional Association for Transgender Health’s (WPATH) Standards of Care, currently in its 8th version.²⁰” (Introduction, Page 4)

The aims do feel a bit far-reaching and disjointed. As I have stated, the early-life environmental exposures and genetic underpinnings and other biological aspects feel odd nestled within this study and warrants a study of its own I feel.

Thank you for your comment. As we tried to clarify above, the SKDS was designed to be a relatively big cohort study aiming to fill knowledge gaps in the field; therefore, it has several aims, even though those are not connected directly. Having said that, our main goal is to deepen our understanding of mental health outcomes, psychosocial outcomes and gender-affirming treatments in accordance with the contemporary needs of persons with gender dysphoria. Thus, we divided the study aims into primary and secondary ones. We hope it reads better now.

“The Swedish Gender Dysphoria Study (Svenska Könsdysforistudien - SKDS) thus primarily aims to investigate (1) mental health outcomes associated with GD, for instance, depressive and anxiety symptoms and suicidality; (2) psychosocial outcomes in relation to GD, such as quality of life, global functioning and social vulnerability (along with support, discrimination and violence); (3) different aspects of GAT, including the impact on mental health and psychosocial outcomes; and secondarily (4) biological correlates of GD containing the effects of early-life environmental exposures, genetic underpinnings, epigenetic alterations and potential prognostic biological markers.” (Introduction, Page 5)

Methods

I feel the information on the qualitative part of the study is very much missing in the methods. I would like to know how the interviews are structured as well as what theoretical stance you are taking and how the data is being processed.

Response: Thank you for this comment. We realized that the part about the qualitative study is beyond the scope of this protocol article, and the information about qualitative interviews might be confusing. The qualitative study is not a part of SKDS, and there is no overlap more than obtaining consent from participants in SKDS to share their contact details for recruitment purposes in the separate qualitative study. Therefore, we removed the information about that part in order not to mislead the readers.

Patient and Public Involvement

Can you define further which interest organisations are taking part? In addition, are any trans people involved in the PPI?

Response: Thank you for your comment. RFSL Ungdom and Transammans, the most well-known interest organizations for Transgender rights in Sweden, have been involved in our study. We

received feedback from persons with gender dysphoria through representatives of those interest organizations. We updated the Patient and Public Involvement part accordingly:

“During the recruitment phase a collaboration was established with representatives from interest organizations in Sweden, which are RFSL Ungdom and Transammans; this collaboration is ongoing with regular meetings. Persons with GD were involved in the design of SKDS through interest organization representatives; the researchers were informed by the representatives concerning the contemporary needs of individuals with GD, and the research questions, methods and outcome measures were adjusted and further developed based on the feedback received.” (Patient and public involvement, Page 8)

Ethics and Dissemination

Can you confirm when the study will be completed by the participants and where? If it is presented to the patient at their initial assessment this could perhaps have an influence on the results, even though the participants are informed it does not form part of their assessment.

Response: Thank you for this comment. The planned follow-up duration for each study participant with gender dysphoria is three years, as it was stated in Study design and participants. Having said that, participants were also informed that they could cancel their participation without any justification during follow-up. We updated the manuscript in line with this:

“Study participants were also informed that they could cancel their participation without any justification at any stage of follow-up.” (Ethics and dissemination, Page 8)

The clinical assessment and participation in SKDS are two separate processes occurring in parallel. Apart from the initial recruitment at the clinics, the collection of survey data occurs through web surveys and is not connected to the clinical assessment in any way.

My main issue with the research is that I feel the biological basis for

Reviewer: 2

Prof. Els Elaut, Ghent University

Comments to the Author:

The current manuscript describes the Swedish Gender Dysphoria Study, an ongoing longitudinal cohort study (since 2016) among participants with gender incongruence over 15 years and older. Recruitment is still taking place in six gender teams in Sweden. Unique for this study is not only a web survey (assessing psychosocial and mental health outcomes and prognostic markers) is used, but also biological and cognitive data are collected. A matched control group was recruited. A secondary aim is to study the underlying mechanisms of gender dysphoria.

The authors have written a very clear and comprehensive study description and are complemented with this very thorough and at the same time broad, biopsychosocial approach. A significant amount of participants could be included, and is expected to increase during the last phase of recruitment.

I only have a few very minor suggestions to improve the manuscript.

Thank you for your positive review of the study protocol and your constructive suggestions.

1/ In the abstract, the authors mention an increased prevalence of mental issues in the limitations. Could they provide a reference for this sentence?

Response: Thank you for your comment. With the sentence you mentioned, we aimed to point out a potential limitation which is that persons with mental health issues might be less willing to participate in the study and this may affect the representativeness. As this sentence is a part of the abstract, we cannot provide a reference for it. To make the sentence more understandable, we rephrased it accordingly:

“The representativeness of the target population could be affected by the fact that persons with more mental health issues might tend to participate less often or dropout in higher extent.” (Abstract, Page 2)

2/ In the meantime, WPATH has released the 8th version of the Standards of Care. Hence, I would suggest to update the reference on p. 6, second paragraph.

Response: We updated the reference as you suggested:

*“Treatment guidelines are provided in the World Professional Association for Transgender Health’s (WPATH) Standards of Care, currently in its 8th version.”*²⁰ (Introduction, Page 4)

3/ On p.8 rule 43, the authors refer to several assessment times: baseline, before gender affirming hormonal treatment, and at least six months after. Could the authors clarify what those six months after mean: six months after the start of hormonal treatment? And why was this at least six months after, what is the maximum time since the start of hormonal treatment upon this assessment time?

Response: Thank you for this comment. We aimed to mean at least six months after start of gender-affirming hormonal treatment and now updated the sentence in line with that. Six-month time period was decided to give enough time to detect potential newly generated cognitive influences due to hormonal treatment. There is no maximum time limit for cognitive assessment of cognitive measures after gender-affirming hormonal treatment; however, this is surely an important factor that needs to be taken into account for future data analyses focusing on this part.

“All cognitive tests are performed at baseline, before GAHT and at least six months after start of GAHT.” (Data collection and measures, Page 7)

4/ On p. 9, the authors describe how they ask the participants to consent to have a relative contacted for another study on the experiences of relatives on processes in transition-related health care. What instructions are being given on the participants to prevent both positive and negative selection bias in this other study?

Response: Thank you for this comment. We realized that the part about the qualitative study is beyond the scope of this protocol article, and the information about qualitative interviews might be confusing. The qualitative study is not a part of SKDS, and there is no overlap more than obtaining consent from participants in SKDS to share their contact details for recruitment purposes in the separate qualitative study. Therefore, we removed the information about that part in order not to mislead the readers.

5/ Also on p.9: could the authors explain the abbreviation 'TCS' when used for the first time?

Response: Thank you for your comment. We explained the abbreviation in the text as it follows:

“Regarding statistical power analysis, we need data for 52 participants of each birth-assigned sex in order to detect a 25% increase from a mean Transgender Congruence Scale (TCS) score of 2.1, with $\alpha=0.05$ and power=90%, using a Mann-Whitney U test for the outcome of GD.” (Statistical Analysis, Page 8)